# 3D Printing of Bioinert Oxide Ceramics for Medical Applications

**DOI:** 10.3390/jfb13030155

**Published:** 2022-09-17

**Authors:** Irene Buj-Corral, Aitor Tejo-Otero

**Affiliations:** Department of Mechanical Engineering, Barcelona School of Industrial Engineering (ETSEIB), Universitat Politècnica de Catalunya, Av. Diagonal 647, 08028 Barcelona, Spain

**Keywords:** additive manufacturing, ceramics, 3D printing, medicine, zirconia, alumina, titania

## Abstract

Three-dimensionally printed metals and polymers have been widely used and studied in medical applications, yet ceramics also require attention. Ceramics are versatile materials thanks to their excellent properties including high mechanical properties and hardness, good thermal and chemical behavior, and appropriate, electrical, and magnetic properties, as well as good biocompatibility. Manufacturing complex ceramic structures employing conventional methods, such as ceramic injection molding, die pressing or machining is extremely challenging. Thus, 3D printing breaks in as an appropriate solution for complex shapes. Amongst the different ceramics, bioinert ceramics appear to be promising because of their physical properties, which, for example, are similar to those of a replaced tissue, with minimal toxic response. In this way, this review focuses on the different medical applications that can be achieved by 3D printing of bioinert ceramics, as well as on the latest advances in the 3D printing of bioinert ceramics. Moreover, an in-depth comparison of the different AM technologies used in ceramics is presented to help choose the appropriate methods depending on the part geometry.

## 1. Introduction

Ceramics are made of inorganic compounds. They are synthesized at high temperatures. They are brittle, hard-to-be machined materials [1]. Bioinert ceramics can be classified into two groups: non-oxides and oxides. Among non-oxides, carbon, diamond-like carbon, titanium nitride, zirconium nitride, and silicon nitride are usually considered. Among oxides, commonly used ones are alumina (Al_2_O_3_), zirconia (ZrO_2_), and titania (TiO_2_) [2]. These oxides exhibit special properties that make them biocompatible. For example, they have high chemical inertness and high biological safety [3], as well as high mechanical strength, low corrosion, and high wear resistance [2]. The main application of zirconia is the manufacture of dental implants, while in association with alumina it is used in different medical applications.

According to the FDA (Federal Drug Agency), medical implants are devices placed inside or on the body’s surface [4]. These devices are regulated to ensure safety and effectiveness for the patient. There are different types of medical implants: (1) some are prosthetics, whose aim is to replace any missing body part; (2) others are used for drug delivery; (3) those used to monitor body functions; and (4) those providing support to both tissues and organs. These devices are products they should be able to be used in almost every organ of the human body [5]. In addition, these medical implants need to satisfy certain requirements to be implanted in the body [2]: (1) corrosion resistance, especially in the case of metals; (2) biocompatibility, so that the tissue does not reject it; (3) bio-adhesion, because the implant needs to be added to the human body; (4) biofunctionality: in other words, the device needs to continue either performing the function of the replaced part or helping it to be carried out properly; (5) the devices need to be processable; and (6) the device should match the mechanical performance of the corresponding tissue.

In order to verify the fulfillment of these requirements, different tests can be performed on the materials used in medical devices. Firstly, from the biological point of view, studies of genotoxicity, cytotoxicity, or sterilization are important. Secondly, physical, and chemical studies are essential to avoid the disintegration of implant material from the human body due to corrosion. This effect might have harmful consequences not only for the tissue where the device is placed but also for the surrounding tissues and organs. 

According to the abovementioned requirements, long-term medical implants could be developed to be permanent, for instance stents or hip implants. However, there are some devices such as chemotherapy ports or screws to repair broken bones that can be removed once they are no longer needed. 

Most of these medical devices can be manufactured using different materials: metals, ceramics, or polymers. Although metals and polymers have been widely used in medical devices, the use of ceramics is starting to bloom. Ceramic materials are used in many different fields such as electronics, automotive, aerospace, and biomedical engineering. Regarding the latter one, ceramics have been used in different applications: dental implants, scaffolds, prostheses (hip and knee), etc. 

Metals are usually employed in applications such as hip or knee implants. Additionally, many metallic parts used in the medical sector have complex shapes, in many cases combined with porous structures that favor their fixation in the body using osseointegration. On the other hand, metals should be biocompatible [6] and have good mechanical properties [7], i.e., high tensile, compressive, and shear strength, high fatigue strength to prevent failure under cyclic loading, and their elastic modulus should be comparable to that of bone. They should also have high corrosion resistance, high wear resistance, and relatively low price. Another important factor to be considered is the possibility to obtain porous structures because they influence both the mechanical strength and the biological properties of the tissues. On the other hand, the osseointegration of bones depends on both biomechanical interlocking and biological interactions, which are related to the surface roughness of the implants [8]. The most common materials are stainless steel [9], titanium [10], and cobalt–chromium alloys [11]. 

In terms of polymers, there are mainly two types: natural and synthetic polymers. Natural polymers, also known as biopolymers, commonly occur in nature and can be extracted, such as cellulose, collagen, gelatine, or alginate. They are used in tissue engineering applications due to their excellent biocompatibility properties. However, as they have low mechanical properties, their target applications are normally soft tissues [12]. Regarding synthetic polymers, they are human-made polymers produced by means of chemical reactions; for example, polyester, polyurethane [13], or nylon, among others. In 3D printing technologies, it is very common to use PLA (the most common polymer material used in FDM 3D printers) for the manufacture of scaffolds, either without any fill or filled with other materials as metals [14]. Additionally, there is the possibility of synthesizing hydrogels, which are hydrophilic gels, polymer networks that are swollen with water as the dispersion medium [15]. 

Ceramics are versatile materials. Their properties include high mechanical strength and hardness, good thermal and chemical stability, appropriate optical, electrical, and magnetic performance as well as good biocompatibility. Nevertheless, machining of ceramics is difficult since the material is fragile and the thermal stresses due to the cutting operation may lead to breakage of the pieces. For this reason, Additive Manufacturing (AM) technologies can be an alternative way to manufacture ceramics. AM technologies are employed to obtain 3D printed physical parts in a point-by-point, line-by-line, or layer-by-layer way from 3D CAD models that are digitally sliced into cross-sections. There are different types of ceramic materials: bioactive ceramics (bioglass, wollastonite, hydroxyapatite (HA) or phosphates), bioinert ceramics (alumina, zirconia and titania) and composite materials. 

Bioactive materials can induce precipitation in contact with body fluids, which is of great interest for their bone bonding capacity [16]. Bioglasses are glass–ceramic materials based on silicon, containing calcium and phosphorous. Bioglass nanoparticles (BGs) have special properties of osteoconductivity and osteoinductivity. Under certain conditions, they have bactericidal properties [17]. Other important ceramic materials in medical applications are wollastonite and hydroxyapatite (HA). On the one hand, wollastonite is a functional filler that has great potential to be used in thermoplastic composites, replacing more expensive reinforcement such as glass fiber [18]. It can be obtained in three different ways [18]: (1) natural wollastonite; (2) synthetic wollastonite; and (3) chemically modified wollastonite. It has high mechanical strength [19] and it is biocompatible [20]. On the other hand, hydroxyapatite (HA) is one of the most studied ceramics for various biomedical applications such as bone tissue engineering or coatings. It has a highly biocompatible behavior. However, it has lower mechanical properties for load bearing applications; lower antimicrobial activity and lower biological interaction rates than other ceramics [21]. Additionally, HA formation is highly dependent on the bioactive glass composition [22]. Other phosphates such as tricalcium phosphate (TCP) have been widely employed in bone regeneration, for example in coatings, in order to improve the bioactivity of the implants. Calcium phosphates are present in human bones [23]. 3D printed calcium phosphate scaffolds have also been used in bone engineering [24,25].

Composite materials are known as a mix of different materials. For example, in hop prostheses it is usual to employ zirconia-toughened alumina (ZTA) in order to increase the toughness of alumina [26].

To the best of the author’s knowledge, few documents are known that summarize the use of 3D printed bioinert ceramics in medical applications. Therefore, this review aims to discuss the state-of-the-art of the use of bioinert ceramics for medical implants using AM technologies. Firstly, a summary of the current 3D printing technologies for ceramics is introduced. This is followed by a description of the bioinert materials that are used in medical implants. Last but not least, different applications are discussed, and different examples are introduced for a better understanding. 

## 2. AM Technologies for Ceramics

There are seven categories of AM technologies according to ISO/ASTM 52900 [27] (see Figure 1): (1) Vat Photopolymerization, (2) Material Extrusion (ME), (3) Material Jetting (MJ), (4) Binder Jetting (BJ), (5) Powder Bed Fusion (PBF), (6) Directed Energy Deposition (DED), and (7) Sheet Lamination. For the manufacturing of ceramics, these seven categories can be employed. Different types of feedstock are used [28]: (1) slurry-based, (2) powder-based, and (3) bulk-solid-based (see Table 1). The different AM processes are presented in the subsequent subsections.

### 2.1. VAT Photopolymerization

Stereolithography (SLA) is a laser technology that uses a liquid resin sensitive to UV light. A UV laser beam scans the surface of the resin and hardens selectively the material corresponding to the cross-section of the product, so that the 3D part is usually created from the bottom up. The necessary supports are generated automatically and are removed manually at the end of the process. This technology was first developed by Hull in 1986 [29]. This technology produces models with an excellent appearance and optimal quality in short time. Additionally, it is possible to manufacture small series of complex geometries and large artistic models. However, there is an issue with the adhesion between the layers during printing: every time a layer is cured it shrinks and, consequently, it leads to residual shear stress between the 3D printed layers. 

DLP or digital light processing is another type of VAT photopolymerization technique whose major difference from SLA is a projection of UV or visible light from a digital projector to flash a single image of the layer at once. For example, 8% mol YSZ (Yttria Stabilized Zirconia) electrolytes have been obtained by means of DLP, using a UV resin slurry [30]. 

An example of a machine that uses the SLA technology is Form 2 of Formlabs. It has the following characteristics: laser focal point diameter of 140 µm, build volume of 145 × 145 × 175 mm^3^, and layer thickness of 25–300 µm [31].

### 2.2. Material Extrusion (ME)

Material extrusion (ME) is a very common 3D printing technique, which is used in a wide range of sectors. Fused Deposition Modelling (FDM), also known as Fused Filament Fabrication (FFF), uses a continuous filament of a thermoplastic material, which is extruded and deposited with the help of a nozzle. In 1989, Scott Crump patented the FDM process, which expired in 2009 [32]. From that point, this technology has grown steadily as an open-source technology. It is normally used with polymers, but it is possible to use ceramic-filled filaments. These filaments are prepared by loading ceramic particles (commonly up to 80 vol%) into thermoplastic binders, so they are composite materials [28]. On the other hand, Direct Ink Writing (DIW), also known as Robocasting (RC), is an AM technology based on the direct extrusion of inks in suspension. It consists of the deposition of highly colloidal suspension ceramic powder concentrates in water, with the addition of an organic compound. These compounds need to have proper rheological properties: (1) relatively lower viscosity under stress than without application of such stress (optimal fluidity through the nozzle); (2) excellent capacity to retain shape after deposition; and (3) a rheological behavior of shear-thinning presenting the ink a pseudo solid feature. Finally, after obtaining the desired rheological behavior, the ink is placed in a syringe and is generally extruded through a conical/cylindrical nozzle.

Within the medical field, it is possible to 3D print, for example, polyamide (PA) with alumina or with zirconia [33]. For instance, with a Sigma R19 3D printer [34], which is provided with the IDEX (Independent Double Extrusion) System, two different filaments can be used to produce a certain part.

### 2.3. Material Jetting (MJ)

Material Jetting (MJ) is based on the spraying of ultra-thin layered photopolymers on a build deck. Each layer of photopolymer cures immediately after spraying with UV light, which allows generating fully cured products that can be handled and used immediately without the need of a post-process phase. The gel-like support, designed to allow the construction of complex geometries, is subsequently removed by jets of water. Regarding the 3D printing of ceramics using this AM technology, it is important to achieve certain rheological characteristics such as dispersity, stability, viscosity, and surface tension. That is why several optimizations and compromises need to be considered out to ensure that proper solid loading and rheological properties are achieved. For example, DMA (Dynamic Mechanical Analysis) in both compression and shear tests are an option. An important drawback of this AM technology is the poor surface finish (high surface roughness), which limits its resolution.

Alumina parts have been manufactured by means of 3D inkjet printing (IJP) using different binders such as polyvynil alcohol (PVA), polyvynil acetate (PVAc) and arabic gum. Lowest porosity was obtained for PVA and arabic gum [35].

An example of a 3D printer based on the material jetting technique is Carmel 1400 [36], which was developed by XJET. It offers high productivity as well a geometric complexity option. It has a build volume of 500 × 280 × 200 mm.

### 2.4. Binder Jetting (BJ)

In 1990, Emmanuel Sachs at the Massachusetts Institute of Technology (MIT), USA, proposed the method of Binder Jetting (BJ) [37]. This method consists of the following steps [38]: (1) preparation of raw materials, which are powder-based, and the pretreatment of powder because the binder and powder bond need to be resistant; (2) 3D printing of the powder into a green body; (3) curing, which is the strengthening of the green body via polymerization or cross-linking; (4) organics are removed by a process known as de-binding, in which the heating rate, holding temperature, and holding time are controlled; and finally, (5) the 3D printed ceramic parts receive a post-process such as CVI (Chemical Vapor Infiltration), PIP (Precursor Infiltration and Pyrolysis) or RMI (Reactive Melt Infiltration). Additionally, a major advantage of binder jetting over the other AM technologies is that the shrinkage due to the sintering process, which appears in conventional ceramic preparation processes, is avoided. For instance, YSZ parts have been obtained by means of binder jetting, using an inorganic colloidal binder [39].

ComeTrue^®^ CERAMIC 3D printer [40] customizes ceramic powders. It has a build volume of 200 × 160 × 150 mm. 

### 2.5. Powder Bed Fusion (PBF)

Within powder bed fusion (PBF), two different types of 3D printing process can be distinguished. On the one hand, SLS (Selective Laser Sintering) was first developed and patented by Carl Deckard and his academic advisor Joe Beaman at the University of Texas at Austin in the mid-1980s [41]. The general principle of operation consists in a laser beam that goes through the surface of the powder material by successively solidifying different layers of material. The material is heated to a temperature slightly below the melting temperature. The solidification occurs through the punctual incidence of a laser beam, which causes heating above the sintering temperature. This occurs when grain viscosity decreases with temperature, causing superficial lesions which, without merging, generate an interfacial union between the grains. Then, dust grains that are not overheated remain unbonded and, consequently, act as supports. Regarding the remaining powder, the material that is not solidified is removed at the end of the manufacturing process. Additionally, two major concerns in the SLS of ceramics are the high shrinkage and the high porosity remaining in the final parts [28].

On the other hand, the SLM (Selective Laser Melting) process was developed and patented at the Fraunhofer Institute for Laser Technology (ILT) in Germany in 1996, DE 19649865 [41]. Then, in 1997, a patent was filed in the USA [42]. This process, in comparison with SLS, uses laser sources with much higher power and it does not require any secondary low-melting binder powder. This AM technology consists of a laser melting the metal particles to join them. The laser processing parameters are crucial to the quality of the fabricated parts [28]. Next, successive layers of powder are spread over each other, while the laser selectively joins particles to build the part and its support. Additionally, the excess dust is removed. Finally, the support is removed, and the 3D printed part is obtained. Moreover, one of the most vital fabrication parameters is the slice thickness, which influences the manufacturing time and the surface roughness of the 3D printed part. For example, smaller slice thickness reduces surface roughness but implies a longer production time, whereas a larger thickness has the contrary effect.

The use of PBF processes is limited for ceramics, because most ceramic powders have low absorptance [43]. However, some authors have worked in this area; for example, Juste et al., who obtained oxide ceramic parts with the SLM technology [44].

The Sinterstation^®^ 2500 system was developed by 3D Systems [45]. It has a build volume of 381 × 330 × 457 mm and includes a Coherent 70 Watt CO_2_ laser. 

### 2.6. Directed Energy Deposition (DED)

DED was laid at Sandia National Laboratories during the beginning of the 1990s. It is a laser additive manufacturing technology that uses thermal energy to fuse materials by melting as they are being deposited, forming a molten pool [46]. After the laser beam moves to another position, the heat begins to dissipate, and the molten materials start to solidify. The first layer provides the subsequent layer fabrication of a new “substrate” for continuing the 3D printing process. A similar process will be repeated layer-upon-layer until the designed 3D structure is completed. For this process, normally four-or five-axis machines are used and, consequently, it is possible to deposit material from any angle onto the existing surface of the 3D printed part [8]. This process offers several advantages over traditional AM technologies because it is not so time-consuming and is not so expensive. Additionally, there is no need for binder materials nor post-sintering process. Directed energy deposition is commonly used for metals. However, metal matrix composites (mainly Ti-based) can be additively manufactured by this technique [47]. 

LENS 450 of Optomec is an example of this technology [45]. It has a build volume of 100 × 100 × 100 mm. The linear resolution is around ±0.025 mm. 

### 2.7. Sheet Lamination

In 1984, Kunieda reported for the first time the sheet lamination technique, which was then further developed by the Helysis Corporation in 1986 [48] and commercialized in 1991. Sheet lamination is a process in which sheets of the material are bonded to form the 3D printed parts. The sheets are joined one after the other by using an adhesive or a laser beam. Next, these sheets are cut with a laser or knife to form the required shape. This process is repeated until the object is formed. An example of this technology for 3D printing and ceramics is Helysis LOM 1015. LOM was used to obtain alumina parts from a green tape which was previously manufactured by the roll-forming technique [49].

## 3. Comparison among the Different AM Technologies for Ceramics

In order to compare the different AM technologies for ceramics, the shape of the obtained parts is considered in the present section. According to Lewis et al. [50] and Feilden [51], the 3D printed parts are classified into different categories depending on their structures (see Figure 2): (1) filling; (2) spanning; (3) overhanging; (4) floating; and (5) closed cavity. Among the 3D printing techniques that have been summarized in Table 1, some have more capability than others to achieve each one of the mentioned features. Table 2 compares the abilities of each 3D printing technique: (1) ‘easy’ to manufacture the ceramic part (green color); (2) ‘difficult’ in the 3D printing of this type of ceramic shape (yellow); and (3) ‘impossible’ to obtain that shape (red). 

In Table 2 it can be observed that the BJ, SLS, SLM and LOM technologies for ceramics allow obtaining all the shapes but the closed cavities.

## 4. Bioinert Ceramics

A biomaterial is a material that is intended to act interfacially with biological systems to evaluate, treat, augment, or replace any tissue, organ, or function of the body. The most important parameter of biomaterials is biocompatibility because it guarantees the success of the application of the biomaterial in the human body. Biocompatibility is defined as the ability of a material to carry out its processes with an appropriate response from the guest in a specific situation. There are three generations of biomaterials [52,53].

The first generation of biomaterials corresponds to the minimum response from the guest. They are used to generate a reaction to a foreign body and are isolated from the environment. The objective of these biomaterials is to achieve an appropriate combination of physical properties equal to those of the replaced tissue with the minimum toxic response. The most typical biomaterials are inert materials. Within ceramics, there are two typical examples: zirconia and alumina. 

The second generation of biomaterials includes those to which the host’s response is inevitable. They are designed to produce specific and beneficial responses, such as adherence and internal growth, among others. There are two types of biomaterials within this generation: (1) absorbable, such as biodegradable polymers, whose chemical degradation and absorption are controlled, while it is replaced by host tissue; and (2) bioactive, such as bioactive glasses or ceramics, which produce a controlled action and reaction in the physiological environment. 

Tissue engineering corresponds to the third generation of biomaterials, in which the interaction between biological entities and synthetic materials is the main point. Normally, within this biomaterials group, products such as scaffolds are mainly designed. After this, cells are seeded, although the scaffold can already be 3D printed with cells in it. In that case, it would be known as a bio-ink. Then, once the scaffold is implanted in the human body it can either be reabsorbed or remodeled actively.

Among all these materials, the present manuscript is focused on bioinert ceramics, which are characterized by their porosity, hardness, low friction, high chemical stability, and high flexural strength. The most common types are zirconia (ZrO_2_) and alumina (Al_2_O_3_). Bioinert ceramics are promising since they have excellent chemical stability, biocompatibility, corrosion restriction behavior, and wear resistance [54]. However, a great disadvantage of these materials is that their mechanical strength diminishes during the reabsorption process [55]. In addition, some in vivo studies showed poor bonding to bone tissue, for example in zirconia ceramics [56].

### 4.1. Zirconia

Zirconia, also known as zirconium dioxide, and sometimes called “cement steel” because of its crystalline reticulation changes when force is applied to the surface, was first identified in 1789 [57]. It was used in the 18th century as a pigment for ceramics [58]. In 1808, Humphry Davy tried to isolate it using electrolysis but failed [59]. Next, in 1824, Berzelius was able to obtain zirconium metal in an impure form by heating a mixture of potassium and potassium zirconium fluoride in an iron tube [60]. Nevertheless, it was not until 1969 that Helmer and Driskell [61] published the first research study about its use in biomedical applications. The zirconia crystals can be arranged in three different patterns [62]: monoclinic (M) at room temperature, cubic (C) above 1170 ºC, and tetragonal (T) above 2370 ºC [63]. In 1972, Garvie and Nicholson found that alloying zirconia with oxides like calcium (CaO), yttria (Y_2_O_3_), and magnesia (MgO) could stabilize the tetragonal phase of zirconia, preventing its transition from the tetragonal to the monoclinic phase and producing ceramics with previously unseen crack resistance [64]. Specifically, nowadays yttria-stabilized zirconia (YSZ) is used to manufacture different kinds of prostheses such as the dental, knee, or hip prostheses [65,66], because of its high fracture toughness and bending strength, higher than those of alumina. The recommended yttria proportion to obtain high mechanical strength is 3% molar [67]. 

Zirconia can be divided into four different categories: (1) FSZ (Fully Stabilized Zirconia), which only contains the stabilized cubic phase, due to the high concentration of dopants; (2) TZP (Tetragonal Zirconia Polycrystals), in which the material is composed of almost 100% tetragonal phase, and generally stabilizes with oxides of yttrium (Y) or Cerium (Ce); (3) PSZ (Partially Stabilized Zirconia), which is formed by large cubic phase grains with tetragonal phase precipitates in their interior and small grains of metastable tetragonal zirconia; and (4) DZC (Dispersed Zirconia Ceramics), in which materials are composed of a tetragonal zirconia dispersion (5–30% by weight) in a ceramic matrix, where the fracture toughness is highly dependent on the transformability of dispersed zirconia.

Zirconia offers several advantages over other materials: (1) significant reduction in the wear rate of implants [68]; (2) excellent biocompatibility [69] and low toxicity [70]; (3) reduction in the risk of toxicity [71]; (4) preclusion of the possibility of corrosion resistance [72]; (5) appropriate mechanical properties for the manufacture of medical devices [69] (compressive strength of about 2000 MPa, bending strength of 900–1200 MPa [57], and fracture toughness of 5–10 MPa·m^1/2^ [73]). 

According to Maziero Volpato et al. [70], during the processing of the zirconia, each stage might have a different effect on the microstructure. In the initial powder, a different yttria content and distribution leads to different phases that may change the microstructure. Then, whitening not only decreases the oxygen vacancies but also modifies the residual stresses. Additionally, both grinding and machining have a direct effect on surface roughness and residual stresses as well as initial monoclinic content. Apart from this, changes in the 3D printing process might also have a direct effect on the properties of the material.

### 4.2. Alumina

Alumina, also known as aluminum oxide (Al_2_O_3_), is a biomaterial that was first discovered in 1821 by the geologist Pierre Berthier [74]. Then, in 1856, Henri-Etienne Sainte-Claire Deville along with his partners at Charles and Alexandre Tissier’s produced the first industrial aluminum. After this, in 1889, Karlo Joseph Bayer, an Austrian chemist, invented a cheap and feasible alumina production. This process is still used nowadays in the industry. It is based on the crush of the bauxite, which is mixed with sodium hydroxide, and seeded with crystals to precipitate aluminum hydroxide. The hydroxide is heated in a kiln to drive off the water and produce a few grades of granular alumina such as activated alumina, smelter-grade alumina, and calcined alumina. 

Alumina has been used in implants since 1933 when Rock issued a German patent for its use in “artificial spare parts” [75]. In the 1960s, Sandhaus and Driskell used alumina in dental implants [76], and Boutin developed an alumina-on-alumina prosthesis for total hip arthroplasty [77]. Alumina is a highly stable oxide. It has different metastable phases which transform into σ-Al_2_O_3_ at temperatures above 1200 °C [78]. 

Alumina has the following characteristics [79,80,81,82]: (1) low electric conductivity; (2) high melting point (approximately 2045 ºC); (3) high strength (310 MPa) and high elastic modulus (around 350 GPa) as well as excellent wear resistance; (4) high density, around a 97%; (5) extreme hardness, 9 on the Mohs hardness scale; (6) high corrosion resistance since it is ceramic; (7) lightweight; (8) toughness; and (9) biocompatibility. 

Despite having excellent properties, the preparation method and prototyping precision of the alumina ceramic core are difficult and problematic [83]. Normally, the ceramic core is prepared through the investment casting method, which limits the fabrication process, including a long production cycle, high cost, and low precision [84]. 

### 4.3. Titania

Discovered in the 18th century [85], titania, or titanium dioxide (TiO_2_) is used in dental and orthopedic implants [86]. It is a kind of functional inorganic material that shows a good biocompatibility, bioactivity, and stability [87], and may be an alternative to commonly used metals such as stainless steel or titanium. These materials, although they have high mechanical strength, can cause bone resorption and stress shielding between implants and surrounding bone tissue once implanted. This might directly affect the rehabilitation of patients. On the other hand, one of its major drawbacks is that it requires modification to enhance osseointegration with the bone.

Its main aim is to be used in bone implants. Chen et al. [1] discovered that the elastic modulus of titania ceramics sintered at different temperatures (from 1000 °C to 1400 °C) is between 1.22 GPa and 6.87 GPa, close to the elastic modulus of high-density cancellous bone that is in a range of 0.28 GPa and 1.86 GPa. 

Other biomedical applications of titania are its use as an antimicrobial agent [88], in toothpaste [89], and in ointments [90]. It has also been used in Alumina–Zirconia–Titania composites for dental applications [91].

## 5. Medical Applications of Additive Manufactured Ceramics Materials

Within the medical field, different devices can be developed using AM technologies with bioinert oxide ceramics. The main ones are: (1) dental implants; (2) hip implants; (3) knee implants; and (4) scaffolds. 

### 5.1. Dental Implants

A dental implant is a medical product designed to replace the missing corn and keep the artificial tooth in place. It is manufactured with biocompatible materials in order not to produce rejection and allow it to be attached to the bone. Then, the implant surface needs to have a coating for increasing its adhesion to the bone. This corresponds to the bonding of the tissue to the implant or biointegration. In the past, hydroxyapatite (HA) has commonly been used as a coating on titanium dental implants. Nowadays, zirconia implants have started to be manufactured. Their integration is faster, as mentioned, because their union is not mechanical as in titanium, but chemical. Additionally, as can be seen in Figure 3B, peri-implant infections can occur with a metallic material. Therefore, using zirconia parts could be a solution, as mentioned by Gahlert et al. [92]. 

For example, Wu et al. [84] manufactured alumina parts using stereolithography from particles with different particle size distributions. It was confirmed that both the combination of a powder with a bimodal particle size distribution and the vacuum debinding process provide an effective way to 3D print ceramics with good performance through SLA. Next, Wu et al. [93] combined SLA with liquid precursor infiltration (a technique capable of incorporating the desired content of exotic elements with controllable distribution) to 3D print alumina-toughened zirconia (ATZ). The 3D-printed ATZ sample showed the lowest aging rate and phase transformation depth compared with 3Y-TZP. In this way, they were not only realizing the complex and ultrafine shape free of molds but also suppressing the low-temperature aging behavior of dental implants. Alumina dental crowns have recently been obtained with the SLA technology by Dehurtevent et al. [94]. They studied three different building orientations (flat-XY, on-edge-ZY, and vertical-ZX) and their influence on the physical and mechanical properties of the crowns. The ZY samples with small number of layers provided the best solution. In another example, Zandinejad et al. [95] used SLA to evaluate the potential of 3D printing of ceramic crowns. It was concluded that both monolithic zirconia vs. bi-layered ATZ showed no significant difference in fracture resistance and both can be used for that purpose. Lian et al. [96] manufactured dental bridges in 3Y-TZP, with a ceramic charge of 40% vol. Wang et al. used SLA to obtain zirconia dental crowns with a photosensitive resin [97]. Recently, Coppola et al. [98] 3D printed alumina–zirconia composites utilizing the DLP-based stereolithography. The samples provided higher mechanical properties than previously 3D printed parts with conventional stereolithography.

On the other hand, zirconia was investigated by Osman et al. [65], who found that it is also possible to 3D print zirconia dental implants using the DLP technology. The results showed that the 3D printed parts had sufficient dimensional accuracy and similar mechanical properties to those of conventionally produced ceramics. Then, Lee et al. [99] 3D printed zirconia dental implants using DIW, and the produced parts proved to be free from delamination and cracks. On the other hand, Anssari Moin et al. [100] prepared zirconia root analogue implants by means of the DLP process.

In recent years, Fused Deposition Modelling (FDM) or Fused Filament Fabrication (FFF) technology has been used to obtain alumina dental implants with infiltered glass [101]. Then, Lee et al. [99] 3D printed zirconia dental implants using DIW, and the produced parts proved to be free from delamination and cracks.

Direct Inkjet Printing (DIP) was used to obtain zirconia dental crowns, using a suspension with 27% vol. zirconia content [102]. The same technology was employed by Özkol et al. [103] to prepare a dental bridge framework made of 3Y-TZP (yttria toughened zirconia), from an aqueous dispersion of 40% vol of ceramic.

### 5.2. Hip Implants

Hip implants are medical devices that aim to restore mobility of the patient and relieve the pain usually associated with a variety of different conditions such as osteoarthrit. is, osteonecrosis, developmental dysplasia of the hip, and femoral neck fracture. The surgery that needs to be carried out is known as Total Hip Replacement (THR). For this, a few things need to be taken into consideration: (1) the implant does not last forever and may need to be replaced eventually; and (2) there are different design features and an appropriate one for the patient must be selected. 

Figure 4 shows the typical shape of the hip prostheses. Overall, stems are composed of metals, whereas femoral heads can be both made of metal or ceramic. The acetabulum can be made of metals, ceramics, or polymers, and so is the insert. It provides a smooth gliding surface [105]. Nowadays, there are four main types of bearings (sites at which the movable parts unite to form the joint) in the hip implants [106]: metal-on-polyethylene (MoP) and ceramic-on-polyethylene (CoP) are “Hard on Soft” bearings; while the metal-on-metal (MoM) and ceramic-on-ceramic (CoC) are “Hard on Hard” bearings. The first articulations were introduced in 1950 and were made of MoM. However, during the last decades, CoC superseded metals and polymer bearings due to their excellent clinical performance: (1) no metal ion release and lower wear rates; (2) better mechanical properties than polymers; (3) better lubrication and low friction. For example, Blakeney et al. [107] demonstrated that larger diameter head (LDH) CoC THAs have excellent functional outcomes at medium-term follow-up, with a very low revision rate and no dislocations. 

Then, Zhu et al. [108] 3D printed hip implants using a 3 mol% yttria partially stabilized zirconium oxide (3Y-ZrO_2_). They combined additive manufacturing technologies with antibacterial nanomodification, and it was concluded that an ideal implant could be developed with precise structure, wear resistance, and effective antibacterial properties. 

It is usual to employ ceramic coatings on titanium alloy implants to improve the corrosion resistance of the implants [109]. As for additive manufacturing processes, Heer et al. [110] studied the effect of additive manufacturing of an alumina coating over a Ti_6_Al_4_V implant. DED is one of the best choices, since it offers good performance to obtain the morphology of the coating. It was concluded that alumina addition increased the wear resistance by around 90%, thus, indicating the potential of this material for articulating the surface of load-bearing implants. 

Titania coatings have been used on PLA (polylactic acid) and PCL (polycaprolactone) bases for biomedical applications, using the FDM 3D printing techniques [111]. The particles of titania improved the stability and mechanical properties of PLA and PCL implants.

### 5.3. Knee Implants

Knee implants (see Figure 5) are medical devices that recreate the surface of the joint (damaged bone and cartilage) with different components. Overall, three different parts might be replaced: (1) the lower part of the femur; (2) the top part of the tibia; and (3) the back surface of the patella. The process for this surgery is known as total knee arthroplasty (TKA). Normally, the materials used for this surgery are metals [112]. For example, in the case of using cobalt–chromium several complications (especially if the cobalt content is higher than 10%) can occur such as infection, loosening, instability, or pain. Additionally, according to the 2016 Australian Arthroplasty Register, more or less 2% of revision TKAs are due to “metal-related pathology” [113]. These metal-made knee implants can have an unnatural, hinge-like motion that puts increased strain on the muscles and ligaments supporting the knee, leading to different problems in the knee. In this way, as happened with THA, it is necessary to use other materials such as bioinert ceramics. The first clinical trials were carried out with short-term results to verify the survivorship: five-year follow-up [114] or ten-year follow-up [113]. Despite achieving excellent results, it is known that it is necessary to certify the efficacy of the knee implant over a longer period. For this reason, Nakamura et al. [113] developed a ceramic tri-condylar implant, which has an alumina ceramic femoral component, to evaluate clinical and radiological outcomes and to assess the long-term durability of a ceramic tri-condylar implant over 15 years. On the other hand, Meier et al. [115] showed the possibility of using the metal-free BPK-S ceramic total knee replacement system, since it proved to be a safe and clinically efficient alternative to metal implants. They used a BPK-S Integration knee implant manufactured by Peter Brehm GmbH, Weisendorf, Germany. With all this in mind, it can be said that ceramics must be a choice, and this could be developed in hand with AM technologies and metallic personalized implants using EBM. 

In addition, the mentioned company implanted the first ceramic knee implant in Belgium by Christian Quintart, MD, Head of the Department of Orthopaedic Surgery and Traumatology, Hospital Jolimont (La Louvière), on 20. February 2019 [116].

Schwarzer et al. [117] developed a submicron alumina suspension to be used in the lithography-based ceramic manufacturing (LCM) process; for example, in the manufacture of implants for unicondylar knee replacements. 

The photosensitive binder jetting technique can be used to manufacture alumina parts which could be used to manufacture knee or hip implants [118]. With this technique, photocurable resins are deposited as a binder onto ceramic powder with the inkjet system.

Recently, the Fused Deposition Modelling (FDM) or Fused Filament Fabrication (FFF) process has been used to print alumina parts with medical applications, using polyethylene/paraffin (PE/PW) and polyethyleneglycol/polyvinylbutyral (PEG/PVB) as binders [101].

On the other hand, alumina powder with silica binder have been used to print molds in which Co–Cr knee prostheses have been cast [119]. The 3D printing technology consisted of “ink-jet printing” of the silica binder into the alumina powder.

### 5.4. Scaffolds

The manufacture of scaffolds for tissue engineering applications has two approaches: (1) the use of bioinks, which are a mix of cells and a base for the ink (mainly polymers such as gelatine or agarose); and (2) the 3D printing of the scaffolds with the subsequent seeding of cells (see Figure 6) [120]. The latter method is the most common procedure for ceramics. For example, Kim et al. [121] 3D printed biomimetic gelatin/HA biocomposites with effective elastic properties and 3D-structural flexibility. Additionally, Stanciuc et al. [110] used material extrusion to 3D print zirconia-toughened scaffolds for hip arthroplasty. The 3D printed ZTA scaffolds exhibited regularly spaced microporous, rough struts, and fully interconnected macroporosity. Moreover, the human primary osteoblasts were homogenously distributed inside 3D-ZTA and showed increased osteogenic marker expression compared to 2D-ZTA control. 

Cesarano et al. [122] developed the robocasting technology, also known as Direct Ink Writing (DIW), in order to produce alumina parts. Later, Ghazanfari et al. [123] modified the technique to which they called Ceramic On-Demand Extrusion (CODE), in the sense that the part was printed inside a tank with mineral oil and obtained high density alumina parts. Buj-Corral et al. [124] used DIW to print zirconia scaffolds, and they analyzed surface roughness, shrinkage, porosity, and mechanical strength of the parts. Peng et al. obtained low porosity 3D printed zirconia parts to be used in prostheses [125]. However, to the authors’ knowledge, no research has been carried out yet on the use of alumina and zirconia for bioinks. Therefore, most of the efforts have been focused on traditional 3D printing with a posterior cell seeding.

## 6. Outlook and Perspectives

The future of bioinert ceramic manufacturing, in terms of 3D printing technologies, could be found in the next two approaches. On the one hand, the FRESH (Freeform Reversible Embedding of Suspended Hydrogels) 3D printing technique can be used. It is based on the extrusion of the materials. For that, it uses a thermo-reversible support bath, also known as the reservoir, which enables the deposition of materials for manufacturing complex structures [126]. On the other hand, bioinert ceramics can be 3D printed with light-based 3D printing technologies. For that, it would be necessary to mix them with photoinitiators to achieve the necessary photo-crosslinking. There is a new technology, known as volumetric 3D printing, which could be interesting for this approach, as it only takes tens of seconds to 3D print centimeter-scale structures [127].

## 7. Conclusions

AM processes are divided into seven categories. All of them allow obtaining ceramic parts. The technologies that are more commonly used to print ceramics are: Stereolithography (SLA), Digital Light Processing (DLP), Fused Deposition Modeling (FDM) or Fused Filament Fabrication (FFF), Direct Ink Writing (DIW), Binder Jetting (BJ), Direct Inkjet Printing (DIP), and Selective Laser Melting (SLM).

Bioinert ceramics such as alumina and zirconia are characterized by high hardness, low friction, high chemical stability, and high flexural strength. The main biomedical applications of the AM ceramic materials are reviewed in this paper: dental implants, hip implants, knee implants, and scaffolds.

Dental implants can be printed in alumina and in zirconia with VAT photopolymerization technologies such as SLA or DLP. In addition, extrusion technologies like FDM or DIW have been used for the same purpose. Furthermore, the material jetting technology, specifically the Direct Inkjet Printing (DIP) technique, has been used to obtain yttria-toughened zirconia implants.

A few authors have obtained 3D printed ceramic hip or knee prostheses. In one study, the SLA technology was employed to print zirconia prostheses. The same process can also be used for alumina. Binder jetting is another promising technique, in which photocurable resins are deposited as a binder onto ceramic powder. In addition, alumina molds have been 3D printed to obtain knee prostheses by means of the investment casting technique. On the other hand, many ceramic coatings have been used on previously 3D printed titanium alloy implants. Direct Ink Writing (DIW) allows obtaining alumina and zirconia scaffolds.

Although different technologies, mainly within the group of VAT polymerization and extrusion techniques, allow 3D printing ceramic parts, few examples are known about 3D printed implants and scaffolds, highlighting that the use of printed ceramics in medicine has a long way to go.

## Figures and Tables

**Figure 1 jfb-13-00155-f001:**
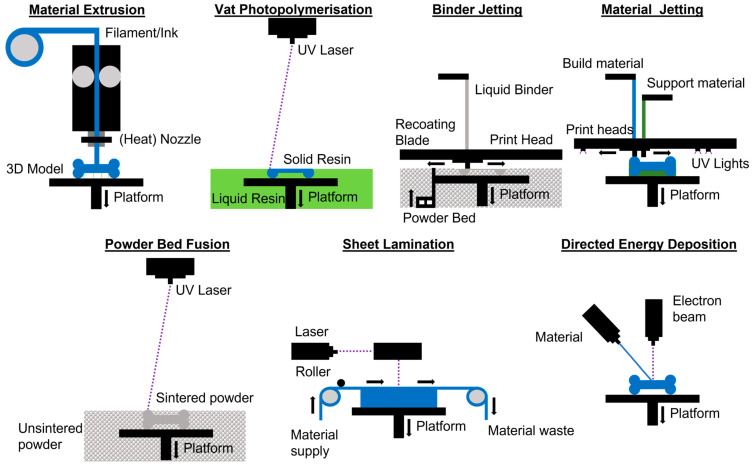
AM technologies scheme.

**Figure 2 jfb-13-00155-f002:**
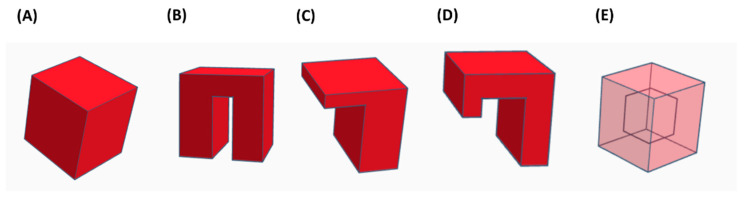
AM technologies scheme. (**A**) Filling. (**B**) Spanning. (**C**) Overhanging. (**D**) Floating. (**E**) Cavity. Adapted from Lewis et al. [22] and Feilden [23].

**Figure 3 jfb-13-00155-f003:**
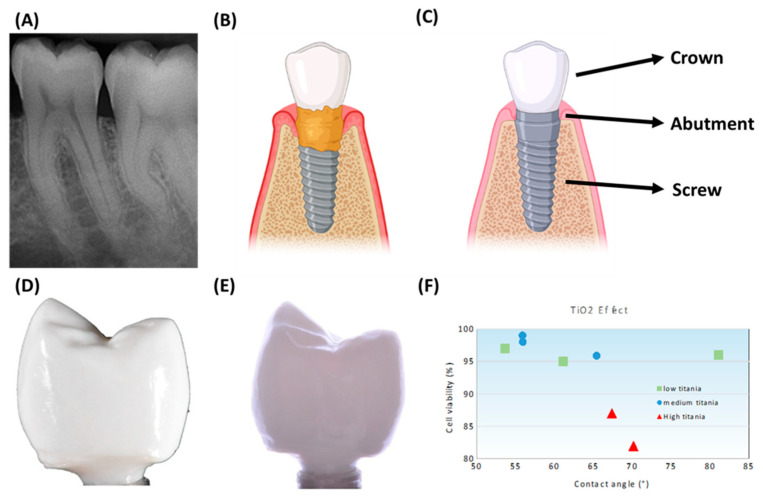
(**A**) Healthy teeth, (**B**) Infected dental implant, (**C**) New dental implant, (**D**) Stereolithography (SLA) additive manufacturing (AM) zirconia crown with buccal marginal defects. Reprinted with permission from [104]. Creative commons CC BY license: https://creativecommons.org/licenses/by/4.0/ (accessed on 5 September 2022). 2019. Merola et Affatato, (**E**) Stereolithography (SLA) additive manufacturing (AM) of implant supported zirconia after cementation. Reprinted with permission from [104]. Creative commons CC BY license: https://creativecommons.org/licenses/by/4.0/ (accessed on 5 September 2022). 2019. Merola et Affatato, (**F**) Variation of cell viability with contact angle of investigated ceramics. Reprinted with permission from [91]. Creative commons CC BY license: https://creativecommons.org/licenses/by/4.0/ (accessed on 5 September 2022). 2020. Khaskhoussi et al. (**C**,**D**) were created from BioRender. The symbol º corresponds to the number of degrees of the angle.

**Figure 4 jfb-13-00155-f004:**
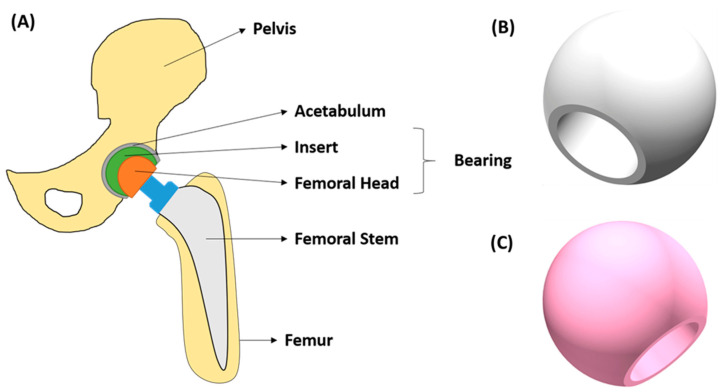
(**A**) Hip implant scheme, (**B**) Zirconia femoral head. Reprinted with permission from [106]. Creative commons CC BY license: https://creativecommons.org/licenses/by/4.0/ (accessed on 5 September 2022). 2021. Khanlar et al., (**C**) Biolox^®^ Delta femoral head. Reprinted with permission from [106]. Creative commons CC BY license: https://creativecommons.org/licenses/by/4.0/ (accessed on 5 September 2022). 2021. Khanlar et al.

**Figure 5 jfb-13-00155-f005:**
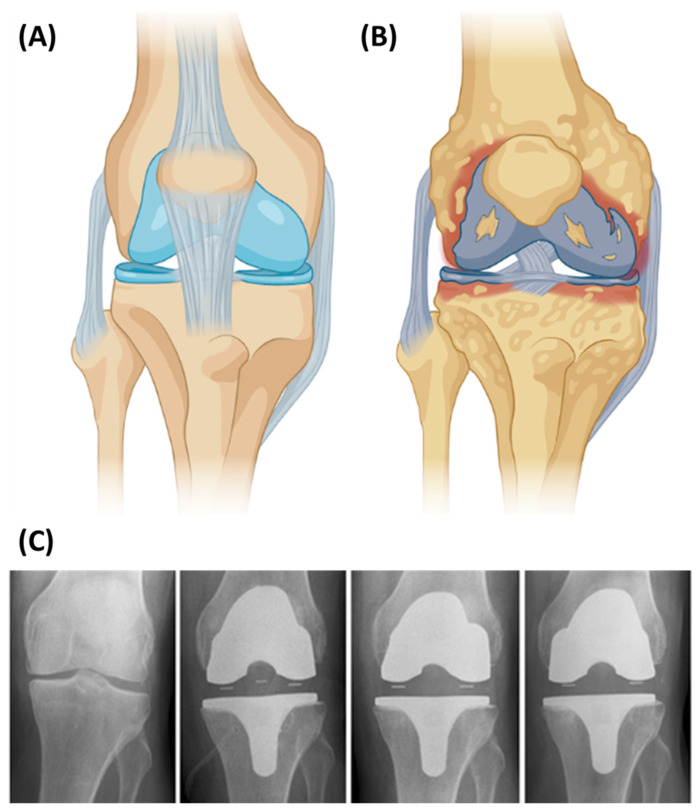
(**A**) Healthy knee, (**B**) Unhealthy knee, (**C**) Radiographs of one patient preoperatively, postoperatively (day of surgery), 3 months and 12 months after surgery. Reprinted with permission from [115]. Creative Commons CC BY license: https://creativecommons.org/licenses/by/4.0/ (accessed on 5 September 2022). 2016. Meier et al. Figure 3A,B were created from BioRender.

**Figure 6 jfb-13-00155-f006:**
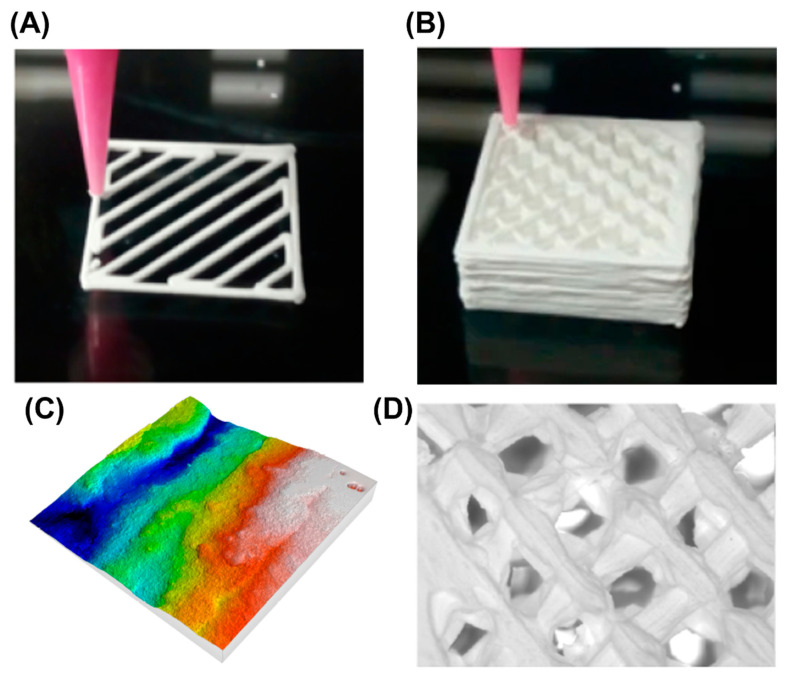
(**A**) 3D printing rectilinear pattern, (**B**) 3D printed zirconia sample. Reprinted with permission from [120]. Creative Commons CC BY license: https://creativecommons.org/licenses/by/4.0/ (accessed on 5 September 2022). 2020. Buj-Corral et al., (**C**) Surface roughness profile of the lateral wall of the 3D printed zirconia sample, (**D**) Microscopic analysis of the 3D printed sample.

**Table 1 jfb-13-00155-t001:** AM technologies.

Feedstock Form	AM Technology	Abbreviation
Slurry-Based	Vat photopolymerization (Stereolithography)	SLA
	Vat photopolymerization (Digital Light Processing)	DLP
	Material extrusion (Direct Ink Writing or Robocasting)	DIW or RC
	Material Jetting (Inkjet Printing)	IJP
	Binder Jetting	BJ
Powder-Based	Powder Bed Fusion (Selective Laser Sintering)	SLS
	Powder Bed Fusion (Selective Laser Melting)	SLM
Bulk-Based	Directed Energy Deposition	DED
	Material Extrusion (Fused Deposition Modelling)	FDM

**Table 2 jfb-13-00155-t002:** Comparison of the ability of each 3D printing technique for ceramics. Green color means ‘easy’, yellow color means ‘difficult’ and red means ‘impossible’.

AM Group	3D Printing Technique	Features				
		Filling	Spanning	Overhanging	Floating	Closed Cavity
VAT Photopolymerization	SLA					
	DLP					
ME	DIW					
	FDM					
MJ	IPJ					
BJ	BJ					
PBF	SLS					
	SLM					
DED	DED					
LOM	LOM					

## Data Availability

Data sharing not applicable.

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
