# Peer review of "3D Printing of Bioinert Oxide Ceramics for Medical Applications"

_jfb, 2022, doi:10.3390/jfb13030155_

Round 1

Reviewer 1 Report

The paper written by Irene Buy-Corral and Aitor Tejo-Otero is focuses on various 3D printing technique and biomedical applications which implies the use of ceramic. The paper is well written and i recommended it publication in Journal of functional biomaterials. However, a minor suggestions needs to be taken into consideration:

- for the figure 1, if the picture is taken from different sources the authors need to include also the copy wright approval.

- subchapter 2.8. i think it could be marked as an individual chapter.  

Author Response

The paper written by Irene Buy-Corral and Aitor Tejo-Otero is focuses on various 3D printing technique and biomedical applications which implies the use of ceramic. The paper is well written and i recommended it publication in Journal of functional biomaterials. However, a minor suggestions needs to be taken into consideration:

Thank you so much for reviewing our paper.

- for the figure 1, if the picture is taken from different sources the authors need to include also the copy wright approval.

The Figure 1 was designed by one of the authors, so copyright approval is not needed.

- subchapter 2.8. i think it could be marked as an individual chapter.  

Subchapter 2.8 was marked as new chapter 3.

Reviewer 2 Report

The paper is devoted to the review of existing techniques for additive manufacturing of ceramic biomedical products.

The technologies SLA, DLP, DIW, IPJ, BJ, SLS, SLM, DED, FDM are considered, the possibility of using each of them for the additive manufacturing of ceramic products is shown by examples, the limitations of the technologies are formulated.

The relevance of the work is not in doubt, the publication is possible in the form provided.

In addition, we would like to wish the authors to continue their work in formulating practical recommendations for the formulation of the process of additive manufacturing of functional ceramic products.

I would also like to draw the authors' attention to the possibility of printing not only with oxide ceramics, but also with wollastonite and hydroxyapatite

10.1016/j.ceramint.2014.09.045, 10.1134/S0036023620020138

Author Response

The paper is devoted to the review of existing techniques for additive manufacturing of ceramic biomedical products.

The technologies SLA, DLP, DIW, IPJ, BJ, SLS, SLM, DED, FDM are considered, the possibility of using each of them for the additive manufacturing of ceramic products is shown by examples, the limitations of the technologies are formulated.

The relevance of the work is not in doubt, the publication is possible in the form provided.

In addition, we would like to wish the authors to continue their work in formulating practical recommendations for the formulation of the process of additive manufacturing of functional ceramic products.

Thank you so much for your kind words.

I would also like to draw the authors' attention to the possibility of printing not only with oxide ceramics, but also with wollastonite and hydroxyapatite

10.1016/j.ceramint.2014.09.045, 10.1134/S0036023620020138

Both references were added as mentioned. Additionally, more references were added as well as important information about wollastonite and hydroxyapatite.

Reviewer 3 Report

This review  is devoted to the 3D of manufacturing of Bioinert Oxide Ceramics for different Medical Applications. This is a very urgent and socially significant problem; therefore, studies in this area are greatly needed.  

The article requires major revision before the consideration for publication in J. Funct. Biomaterials. The following issues need to be addressed:

1. Please describe more clearly the main types of the printed materials used for medical purposes such as metals, polymers, bioactive materials (bioglass, hydroxyapatite and phosphates), composite materials, and bioinert ceramics in the Introduction. Preferably, to summarize their main features in a table and to provide references to relevant literature.

2. The authors should more clearly state why ceramic bioinert materials are promising, what their main advantages and disadvantages are.

3. Please, add the Outlook and Perspectives Section before the Conclusions, ptoviding the main issues and future directions for research to address them. Also, the authors should discuss the unexplored areas of application of Bioinert Oxide Ceramics and new 3D printing technologies to produce materials therefrom.

Author Response

This review is devoted to the 3D of manufacturing of Bioinert Oxide Ceramics for different Medical Applications. This is a very urgent and socially significant problem; therefore, studies in this area are greatly needed.  

Thank you so much for your comments.

The article requires major revision before the consideration for publication in J. Funct. Biomaterials. The following issues need to be addressed:

  1. Please describe more clearly the main types of the printed materials used for medical purposes such as metals, polymers, bioactive materials (bioglass, hydroxyapatite and phosphates), composite materials, and bioinert ceramics in the Introduction. Preferably, to summarize their main features in a table and to provide references to relevant literature.

Introduction. Lines 66 to 90 and lines 106 to 132. More information about other materials was added as suggested.

  1. The authors should more clearly state why ceramic bioinert materials are promising, what their main advantages and disadvantages are.

Section 4. Lines 343 to 347. This information was added as suggested.

  1. Please, add the Outlook and Perspectives Section before the Conclusions, ptoviding the main issues and future directions for research to address them. Also, the authors should discuss the unexplored areas of application of Bioinert Oxide Ceramics and new 3D printing technologies to produce materials therefrom.

New section 6 was added with the Outlook and Perspectives as suggested.

Reviewer 4 Report

The review is ready for publication after minor review.

Author Response

The review is ready for publication after minor review.

Thank you so much for reviewing our paper. Some modifications were carried to the paper (highlighted in red).

Round 2

Reviewer 3 Report

The article can be accepted  in the present form.